# Surgical Treatment of Nonmineralized Supraspinatus Tendinopathy in Dogs: A Retrospective Long-Term Follow-Up

**DOI:** 10.3390/ani13040592

**Published:** 2023-02-08

**Authors:** Lisa Adele Piras, Matteo Olimpo, Pilar Lafuente, Anna Tomba, Sara Del Magno, Elena Lardone, Bruno Peirone, Davide Mancusi

**Affiliations:** 1Department of Veterinary Science, University of Turin, 10095 Grugliasco, Italy; 2Facultad de Salud, Universidad International de La Rioja (UNIR), 26006 Logroño, Spain; 3Norad Diagnostica, 21017 Samarate, Italy; 4Department of Veterinary Medical Sciences, University of Bologna, 40126 Bologna, Italy

**Keywords:** supraspinatus, tendinopathy, arthroscopy

## Abstract

**Simple Summary:**

Tendinopathy is a broad term encompassing painful conditions occurring in and around tendons in response to overuse. The supraspinatus muscle is an important passive stabilizer of the shoulder joint, and it is responsible for shoulder extension and advancement of the limb. Although the cause of supraspinatus tendinopathy is unclear, suggested contributing factors include aging, overuse, chronic trauma and hypoxia secondary to hypovascularity of the supraspinatus tendon. Supraspinatus tendinopathy is a poorly described cause of shoulder lameness in dogs. Two forms of supraspinatus tendinopathy have been reported: mineralized and non-mineralized. While the diagnoses and the treatments of the mineralized form have been widely described in the literature, few scientific papers report the diagnoses, the treatments and the outcomes of the non-mineralized form. This retrospective study aimed to describe clinical, diagnostic imaging, surgical procedure, concomitant shoulder lesions and short and long-term follow-up of dogs with surgically treated non-mineralized supraspinatus tendinopathy. Obtained results indicate a good outcome with a low incidence of complication rate of the surgical treatment of the non-mineralized form of the supraspinatus tendinopathy.

**Abstract:**

(1) Background: two forms of supraspinatus tendinopathy (ST) have been reported in dogs: mineralized and non-mineralized. Surgical treatment consists of longitudinal incisions (splitting) in the tendon of insertion of the supraspinatus muscle. The purpose of this retrospective study is to describe the diagnostic workout, the surgical procedure and the short and long term follow up of dogs treated for non-mineralized ST. (2) Methods: medical records (2010–2017) of dogs diagnosed with non-mineralized ST that underwent surgical treatment were reviewed. Data retrieved were: signalment, history, clinical signs, orthopaedic examination findings, diagnostic imaging findings, surgical treatment, histopathologic diagnosis and clinical outcome. (3) Results: A total of 27 dogs met the inclusion criteria. The most consistent clinical findings were intermittent lameness accompanied by pain on palpation of the insertion of the supraspinatus. Magnetic resonance imaging (MRI) of 27 shoulders distended sheaths of the biceps tendon (10/27), compression of the biceps brachii tendon sheaths (5/27) and enlargement of the supraspinatus tendon (3/27) were observed. The most prominent histologic finding was severe myxomatous degeneration in all 27 samples. Resolution of lameness was achieved in 80% of the cases surgically treated without any further lameness episodes in the long-term follow-up. (4) Conclusions: the surgical splitting of the non-mineralized supraspinatus tendon is an effective procedure with no intra-operative complications and a low incidence of minor (18%) and major (4%) complications.

## 1. Introduction

Tendinopathy is a broad term encompassing painful conditions occurring in and around the tendons in response to injury, especially overuse. The supraspinatus is an important passive stabilizer of the shoulder joint, and it is responsible for shoulder extension and advancement of the limb [1,2]. Although the cause of supraspinatus tendinopathy is unclear, suggested contributing factors include aging, overuse, chronic trauma and hypoxia secondary to hypovascularity of the supraspinatus tendon [3,4,5,6,7]. Two forms of supraspinatus tendinopathy have been reported in dogs: mineralized and nonmineralized [7,8]. Mineralized supraspinatus tendinopathy generally affects large breeds of dog, most commonly Labrador Retrievers and Rottweilers [1,3,7,9]. No sex predisposition has been reported, and the left thoracic limb is more commonly affected [4]. Although dogs with mineralized supraspinatus tendinopathy had higher body weight, no other differences were identified among groups, suggesting that mineralized and nonmineralized supraspinatus tendinopathy are similar entities [7]. Medical treatment includes the restriction of physical activity, anti-inflammatory medication, physiotherapy and rehabilitation, ultrasound therapy, extracorporeal shock wave therapy and regenerative medicine, especially a combination of mesenchymal stem cells (MSCs) and platelet rich plasma (PRP) [10,11,12,13,14,15]. Surgical treatment is recommended when conservative treatment fails and consists of removal of the mineralized foci, longitudinal incisions or splitting of the tendon of insertion of the supraspinatus muscle, and decompression of the biceps tendon, which is frequently impinged by the enlarged supraspinatus tendon [4,16,17,18]. Lafuente and colleagues reported the treatment of nonmineralized supraspinatus tendinopathy, where the part of the supraspinatus tendon in direct contact with the biceps tendon was resected and full-thickness longitudinal incisions were performed along the tendon of insertion of the supraspinatus muscle [7]. While there is a great deal of literature regarding the presentation, clinical findings and diagnostic imaging characteristics of supraspinatus tendinopathy in humans and mineralized supraspinatus tendinopathy in dogs, to the authors’ knowledge, there is limited literature describing the surgical treatment and clinical outcomes in dogs affected by nonmineralized supraspinatus tendinopathy [7,19]. The purpose of this retrospective study was to describe the clinical presentation, diagnostic imaging, surgical procedure, concomitant shoulder lesions, and short- and long-term follow-up of dogs with surgically treated nonmineralized supraspinatus tendinopathy.

## 2. Materials and Methods

### 2.1. Inclusion Criteria

Medical records (2010–2017) of dogs diagnosed with nonmineralized supraspinatus tendinopathy that underwent surgical treatment were reviewed. All dogs at the time of presentation had mono-lateral forelimb lameness and were operated only on the clinically affected limb. Dogs without diagnostic imaging confirmation of supraspinatus tendinopathy, dogs with concurrent orthopedic disorders of the affected limb, and dogs with supraspinatus tendinopathy that were treated conservatively were excluded from the study.

### 2.2. Data Collection

The data retrieved were signalment, history, clinical signs, the findings of the orthopedic examination, the findings of diagnostic imaging, surgical treatment, histopathologic diagnosis and the clinical outcome.

### 2.3. Lameness Evaluation

The disease history was obtained by interviews with the owner, documenting time of onset, the characteristics of lameness, progression of clinical signs and the response to previous treatments.

Complete physical and orthopedic examinations of all limbs were performed, including flexion and extension of the shoulder joint, the biceps tendon test in the German-speaking literature and deep palpation of the supraspinatus and biceps tendons after gait evaluation. Lameness was graded using a scale (0–5) where 0 indicated no lameness, 1 indicated slight lameness only after exercise, 2 indicated slight lameness without exercise, 3 indicated moderate lameness, 4 indicated severe lameness and intermittent non weight-bearing lameness, and 5 indicated non weight-bearing lameness [20].

Pain evoked during the execution of passive movements of the shoulder joint were subjectively graded as 0, no pain; 1, faint; 2, mild; 3, severe. If conscious proprioceptive deficits were noted, a complete neurologic examination was performed, and the case was excluded from the study.

### 2.4. Diagnostic Imaging

Mediolateral and cranioproximal–craniodistal radiographic projections of both shoulders were obtained. During the same anesthetic episode, a synovial fluid sample was collected in cases affected by severe lameness in order to rule out inflammatory pathologies. Magnetic resonance images of both shoulders were acquired with a 0.23 T open unit (MrV by Paramed, Italy), using the following sequences: FSE T2 and GFE STIR T2 on the sagittal and transverse planes, GFE T2 STIR on the dorsal plane and SE T1 on the sagittal plane. Images were obtained from the midscapula to midhumeral regions. The sagittal images were oriented parallel to the craniocaudal axis of the scapula and humerus, and the transverse images were oriented perpendicular to the axis of the biceps brachii tendon, proximal to the intertubercular groove. In order to avoid “magic angle” artifact all shoulders were positioned in a surface coil in slight extension, as described by Agnello [21]. MRI criteria for supraspinatus tendinopathy diagnoses were a marked hyperintense signal in STIR and/or FSE T2 of the tendon [7,22] with loss of its trilaminar appearance [23] and tendon enlargement [22] with or without evidence of impingement of biceps brachii tendon [8]. Other MRI findings such as edema at the myotendinous junction or greater tubercle of the humerus and/or evidence of tendon avulsion and other evidence of joint degeneration increased the index of suspicion for ST tendinopathy [7,8,23,24].

### 2.5. Surgical Technique

Surgery was performed only in the clinically affected shoulder. Dogs were premedicated with methadone^a^ at 0.15 mg/kg IM. Anesthesia was induced with propofol^b^ at 2 mg/kg IV and maintained with isoflurane^c^, aiming for a 0.8% minimal alveolar concentration after endotracheal intubation. Analgesia was provided by a target-controlled infusion of fentanyl^d^ at a plasma concentration of 1.2–1.6 ng/mL. Cefazolin sodium^e^ was administered intravenously (20 mg/kg) 30 min before surgery.

Dogs were positioned in lateral recumbency with the affected limb uppermost. An arthroscopic portal was established cranial and distal to the acromium, and an egress needle was positioned cranially into the joint. A short arthroscope (2.3 mm, 30°) was introduced into the joint through a 2.4 mm sleeve. Joint distention was maintained by an infusion of lactated Ringer’s solution with a lavage pump at 60 mmHg [25]. The medial and lateral glenohumeral ligament, the subscapularis tendon, the articular cartilage of the glenoid cavity, the humeral head and intertubercular groove, the synovium of the gleno-humeral joint, the biceps brachii tendon sheath, and the biceps brachii tendon and its insertion on the supraglenoid tubercle were identified and their integrity was assessed. Biceps lesions were graded according to the classification system proposed by Bardet [26] and colleagues as one of six subtypes: 1, complete or partial avulsion tearing from the supraglenoid tubercle; 2, midsubstance tearing; 3, tendinitis; 4, bipartite tendon; 5, luxation; 6, tenosynovitis of the biceps tendon. After arthroscopy, a craniomedial approach to the supraspinatus tendon and intertubercular groove was performed [25,26,27]. A 3–4 cm longitudinal skin incision was made over the greater tubercle of the humerus. The brachiocephalicus muscle was medially retracted to expose the underlying supraspinatus tendon. A longitudinal incision cranial to the supraspinatus tendon and the greater tubercle in a proximo-distal direction was made to elevate the pectoral muscles and expose the intertubercular groove and biceps tendon. The supraspinatus tendon was thoroughly digitally palpated and inspected for color and consistency, then the medial aspect of the tendon in contact with the biceps was resected and full-thickness longitudinal incisions were performed along the tendon of insertion of the supraspinatus muscle, as previously described [7]. The longitudinal defects in the supraspinatus tendon were not sutured and apposition of the pectoral muscle was performed with interrupted 2.0 or 0 polydioxanone before wound closure. The resected tissue was fixed in neutral-buffered 10% formalin for histologic examination. Any potential intra-operative complication was recorded.

Postoperative pain control was achieved by cold compression for 5 min 3–4 times a day for 48 h, and nonsteroidal anti-inflammatory drugs meloxicam^f^ (0.1 mg/kg SID, PO) were administered for 7 days. No antibiotics drugs were administered in the post-operative period. Dogs were discharged the day after surgery, with instructions to provide strictly restricted physical activity (limited to leash walks of 5 to 10 minutes’ duration four times daily) and passive range-of-motion exercises of both shoulder joints (at least 10 min three times daily) for 3 weeks. During the subsequent 3 weeks, the owners were instructed to gradually increase the duration of the leash walks by 5 min per week. The need of physiotherapy was discussed with the owner and proposed if the clinical outcome was not judged satisfactory.

### 2.6. Outcomes

All dogs were clinically re-evaluated by one of the authors (XXX) 10, 30 and 60 days after surgery. The lameness evaluation score was compared between the entire population and different subgroups affected by only ST or ST with concomitant biceps brachii tendinopathy or ST with concomitant subscapularis tendon partial rupture. Complications were recorded and were either classified as major or minor [28]. Long-term follow-up was performed both by telephone interviews with the owners at least 12 months postoperatively or during a clinical consultation [28]. If the owners could not bring their dogs for a long-term follow-up visit, the functional outcome was evaluated through a telephone interview. The owners were asked to grade their dog’s ability to walk, rise from a lying position, run, climb stairs, play or exercise on a six-point scale: 1, normal, with no difficulties during activity; 2, near normal, with occasional mild difficulty when performing the activity; 3, mildly abnormal, frequently with mild difficulty when performing the activity; 4, moderately abnormal, showing obvious and consistent difficulty when performing the activity; 5, severely abnormal and unable to perform this activity at all; 6, cannot answer, as the activity was not observed or could not be accurately assessed. The overall score was reached by simply summing the scores received for individual questions. This resulted in a continuous score ranging from 5 (least severe symptoms) to 30 (most severe symptoms). The functional outcome was judged as excellent if the score was <6, good at 7 < x < 13, mild at 14 < x < 20 and poor at >20 [29].

### 2.7. Force Plate Gait Analysis

Computer-assisted force platform gait analysis was performed using two force platforms mounted in series (BTS P-6000, BTS Bioengineering, Garbagnate Milanese, Italy) embedded in an 8 m walkway. Two cameras (BTS VIXTA, BTS Bioengineering, Garbagnate Milanese, Italy) connected to the acquisition software (3DGIVEC, BTS Bioengineering, Garbagnate Milanese, Italy) were used to record each trial. The dogs were weighed on a calibrated scale immediately before collecting the force plate data. Each dog was allowed to acclimate to the room before data collection began. The dogs were walked across the force platforms until they appeared comfortable. The walking velocity and acceleration parameters were restricted to ranges of 1.1 to 1.3 m/s and ±0.5 m/s^2^, respectively. The velocity of each trial was measured by three photoelectric cells mounted 1 m apart on the force plate runway that were connected to a millisecond timer in a start–interrupt fashion. The dogs were walked over the force plates by a trained assistant, and the data were collected independently from the left and right side of the body. A valid trial consisted of a forelimb strike, with the complete foot striking the center of the plate and without another foot being on the plate at the same time, followed by an ipsilateral hind foot strike in the same fashion. A single observer (XX) evaluated each foot strike and determined whether a trial was valid or not by means of camera recordings and morphological evaluation of the curve. The trial was discarded if the paw hit the edge of the force plate, if the contralateral paw hit the force plate or if the dog was distracted during the measurements. The peak vertical force (PVF) from five footfalls for each limb was collected at a walk at all time points. The peak vertical force (PVF) was expressed as a percentage of body weight (%BW). Additionally, a corresponding symmetry index (SI) for the hind limbs was calculated for the abovementioned parameters. The symmetry index was calculated using the following formula [30]
SI (%) = [(FR − FL)/(FR + FL)] × 200.
where SI is the symmetry index of the corresponding parameter (PVF), FR is the parameter of the right limb (%BW of PVF) and FL is the parameter of the left limb (%BW of PVF). The SI was expressed as an absolute value.

## 3. Results

Twenty-seven dogs with nonmineralized (NM) supraspinatus tendinopathy met the inclusion criteria. The breeds included were Boxer (five), American Staffordshire Bull Terrier (four), Beagle (two), Labrador Retriever (two), Pitbull (two), Border Collies (two), Argentine Dogo (one), Rhodesian Ridgeback (one), Black Russian Terrier (one), Central Asian Shepherd Dog (one), Pyrenean Mastiff (one), German Shepherd (one), Samoyed (one), Tibetan Mastiff (one), Great Swiss Cowherd (one) and mixed breed (one).

The patients’ ages ranged from 12 to 90 months (mean: 39.7 months). The patients included 18 males (1 neutered and 17 intact) and 9 females (7 spayed and 2 intact). The patients’ bodyweight ranged from 16 to 75 kg (mean: 33.5 kg).

### 3.1. Findings of the Orthopedic Evaluation

All dogs had unilateral thoracic limb lameness. The distribution of the affected limbs distribution was 16 on the right and 11 on the left. Intermittent or waxing–waning lameness was the admitting complaint for all dogs. The median lameness grade evaluated in the entire population was 3 (3.19 ± 0.62). The median duration of clinical signs was 4.7 months (range: 1.5–18 months). All dogs were previously treated with rest and administration of NSAIDs or corticosteroids for 7 to 14 days, without clinical improvement. Pain on flexion of the shoulder and on palpation of the insertion of the supraspinatus tendon was the most consistent finding during the orthopedic examination, occurring in every dog. Median pain grade was two (2.07 ± 0.55). The biceps tendon test in the German-speaking literature was subjectively graded as positive in 3 out of 27 and negative in 24 out of 27.

### 3.2. Imaging

The radiographs showed that 24 out of 27 pathologic shoulders had no radiographic abnormalities, 2 out of 27 had signs of slight shoulder osteoarthritis characterized by the presence of osteophytes in the caudal portion of the humeral head, 1 out of 27 had a deficit of the bicipital groove filling according to a positive-contrast radiographic study. Synovial fluid examination was performed in 16% of the cases and the results of the cytologic examination were compatible with degenerative joint disease.

During the MRI evaluation, a marked hyperintense signal localized to the insertion of the supraspinatus muscle was found in all of the examined shoulders in the STIR images (Figure 1).

The tendon of the supraspinatus muscle was severely enlarged in 3 out of 27 of the cases. Five MRI images revealed compression of the biceps brachii tendon sheaths, while 10 out of 27 of the biceps brachii tendon sheaths were distended. Moreover, an effusion was present in the sheath of the tendon of the infraspinatus muscle in 1 one out of 27 of the cases and in the sheath of the biceps brachii muscle in 1 out of 27 of the cases.

MRI abnormalities of the tendon of the subscapularis muscle were detected in one dog. No MRI abnormalities of the medial and lateral glenohumeral ligament were detected.

Signs of nonmineralized supraspinatus tendinopathy were present on the contralateral forelimb in 10 out of 27 dogs.

### 3.3. Surgical Findings

In all the dogs, surgery was performed by the same surgeon only in the shoulder affected by lameness. Intra-articular abnormalities were arthroscopically observed in 13 out of 27 shoulders and consisted of fibrillation of the medial glenohumeral ligament in 3 out of 27 cases, partial rupture of the tendon of the subscapularis muscle in 7 out of 27 cases and laxity of the tendon of the subscapularis muscle in 3 out of 27 cases (Figure 2).

The arthroscopic and MRI findings were in agreement about the absence of intra-articular shoulders abnormalities in 10 out of 27 shoulders. In 13 out of 27 shoulders, the ectasia of biceps brachii tendon was detected with MRI but was not recognized with arthroscopy. In 9 out of 27 shoulders, lesions of the subscapularis tendon, not previously established with MRI, were diagnosed by arthroscopy. In 3 out of 27 shoulders, laxity of the medial glenohumeral ligament, not previously diagnosed with MRI, was observed by arthroscopy. In 27 out of 27 shoulders, both MRI and arthroscopy were in agreement about the lateral glenohumeral ligament integrity. No intra-operative complications were reported. No dogs required physical rehabilitation in the post-operative period.

### 3.4. Histopathological Findings

A biopsy sample of the supraspinatus tendon from all 27 dogs was submitted for histological evaluation. The sample was collected from the area of maximum thickness of the tendon, at the musculotendinous junction. The histological examination of the tissue specimens revealed discontinuous and disorganized collagen fibers that lacked reflectivity under polarized light in all specimens. The most prominent finding was severe myxomatous degeneration, with the remaining collagen separated by a palely basophilic myxoid matrix and edema.

### 3.5. Short Term Follow-Up

On Day 10, 21 out of 27 cases showed a lower lameness score compared with the pre-operative values. Lameness was classified subjectively as Grade II in 21 out of 27 cases, Grade III in 5 out of 27 cases and Grade IV in 1 out of 27 cases. There was a lack of improvement in lameness in 6 out of 27 cases: in 4 out of 27 cases, the lameness remained the same, while worsening lameness from Grade III to Grade IV and from Grade II to Grade III was observed in 2 out of 27 cases. A fluctuant mass the size of a golf ball was detected under the healed incision in 5 out of 27 cases (Figure 3). It was presumed to be a seroma and was resolved through the administration of NSAIDS and rest for 10 days. In one dog, a recurrence of the seroma was reported on Day 20 and on Day 90, postoperatively.

On Day 30, lameness was graded as absent in 7 out of 27 cases, as Grade I in 5 out of 27, as Grade II in 11 out of 27 cases, as Grade III in 3 out of 27 and Grade IV in one dog. Dogs affected by co-existing supraspinatus tendinopathy and biceps brachii tendinopathy (T30 = 1.6 ± 0.9) showed longer functional recovery, characterized by higher mean values of lameness score, compared to the entire population (T30 = 0.74 ± 1.3), to dogs affected only by ST tendinopathy (T30 = 0.63 ± 1.4) or dogs affected by ST tendinopathy and concomitant subscapular tendon lesion (T30 = 1 ± 0.8) (Figure 4).

The clinical outcome was judged to be unsatisfactory in the dog affected by Grade IV lameness, revision surgery was discussed with the owner, and supraspinatus tenectomy was performed.

On Day 60, 17 out of 27 patients did not show any sign of lameness, 3 out of 27 still had Grade I lameness, 5 out of 27 had Grade II lameness, 1 out of 27 had Grade III lameness and I out of 27 had Grade IV lameness.

### 3.6. Long-Term Follow-Up

All owners agreed to a telephone interview but only 13 out of 27 agreed to a clinical consultation, radiographic study of the shoulder joint and force plate analysis.

Of the 13 dogs that had a long-term clinical evaluation, all had a complete range of motion of the joint and did not show any sign of pain during flexion or extension of the joint or under palpation of the supraspinatus tendon. The symmetry index was between 0.01 and 14.34, with a mean value of 3.19.

A radiographic recheck of the surgically treated shoulder showed mild mineralization in 4 out of 13 dogs and mild degenerative joint disease in 2 out of 13 cases.

Twenty-three owners declared that they were satisfied with the recovery of their dog after surgery, 2 out of 27 were mostly satisfied and 2 out of 27 remained unsatisfied.

Twenty-two of the dogs did not show any sign of lameness on the treated limb, but 9 out of 27 showed some episodes of lameness, which were classified as mild.

When the owners were asked if their pets presented any episodes of lameness on the other forelimb, 25 out of 27 owners answered that no alteration in the gait was noted in that limb.

Gait was judged by the owners as normal in 19 out of 27 dogs, 3 out of 27 had a nearly normal gait, 3 out of 27 had slight but frequent difficulties, and in 2 out of 27 dogs, the difficulty was evident.

The ability to stand up from a sitting position was normal in 12 out of 27 dogs, 8 out of 27 had slight and infrequent difficulties, 5 out of 27 had slight but frequent difficulties, and discomfort was evident in only one dog.

The ability to run was normal in 19 out of 27 dogs, 3 out of 27 had slight and infrequent difficulties, 2 out of 27 had slight but frequent difficulties and 3 out of 27 owners were unable to answer this question.

Nineteen of the dogs did not have difficulty during intense activity and play, 5 out of 27 had slight and infrequent difficulties, and only 3 out of 27 dogs had slight but frequent difficulties.

Twenty-one dogs did not have any difficulty going downstairs and 5 out of 27 had slight and infrequent difficulties; one owner was not able to answer.

After we added the scores of the different activities evaluated by the owners, the functional recovery was judged to be very good in 15 out of 27 dogs, good in 8 out of 27 and sufficient in 4 out of 27 dogs.

## 4. Discussion

In the literature, a breed-related predisposition to supraspinatus tendinopathy has been reported in Rottweilers and Labrador Retrievers [1,3,7,9]. In the present retrospective study, only 7% of the affected dogs were represented by Labrador Retrievers, and no Rottweilers presented at our institution for nonmineralized supraspinatus tendinopathy. The breeds most represented in our population were Boxers and American Staffordshire Bull Terriers. We can argue that a breed-oriented approach (BOA) is not helpful for a diagnosis of the nonmineralized form of the supraspinatus tendinopathy because a clear breed predisposition has still not been proved.

Lafuente and colleagues [7] found a difference in the body weight of dogs affected by a nonmineralized supraspinatus tendinopathy (mean weight: 35 kg) and mineralized supraspinatus tendinopathy (mean weight: 26.2 kg). Our results did not confirm these findings because, in our population, the mean weight of the dogs affected by the nonmineralized form of supraspinatus tendinopathy was 33.5 kg. We can suppose that body weight is not a predisposing factor for this pathology, but it is typical of a medium and large dog, because it has never been described in a small or toy breed of dog [1].

The dogs included in this study were affected by chronic lameness that was intermittent and worsened after physical activity. NSAIDs had been administered previously to all dogs, in association with steroids in some cases, for 7 to 14 days without clinical improvement. The cause of this nonresponse was associated with the degeneration and metaplasia of the tendon, without inflammation [7,31]. In our population, histopathological examination of a portion of the tendon was performed in each case. In all of the samples, the histopathologic changes observed were myxomatous degeneration and chondroid metaplasia, which are characteristics of tendon degeneration, without tissue inflammation. During the physical examination, most dogs included were affected by a severe chronic lameness secondary to shoulder pain. Dogs affected by the nonmineralized supraspinatus tendinopathy included in our study presented lameness characterized by a marked abduction of the limb during the swing phase. We suppose that this lameness can be explained by an attempt by the dog to reduce the pain evoked by compression of the biceps tendon from the impingement of the pathologic supraspinatus tendon [32] during extension of the shoulder. Affected dogs aim to advance the limb, limiting the extension and abducting the shoulder in order to decrease the pain. In our experience, this type of lameness is characteristic of supraspinatus tendinopathy and can add the suspect of the pathology.

The biceps test (in the German-speaking literature) was performed, and it was judged to be positive in 3 out of 27 cases but partial or complete rupture of the biceps brachii tendon was not subsequently confirmed neither by MRI nor arthroscopy. This can be explained by the fact that the biceps test is an extremely subjective evaluation, and the response can be operator-dependent. Moreover, many of the dogs included in our study had a straight poise of the forelimbs, so the flexion of the shoulder determined the extension of the elbow, even if the biceps tendon was intact. This can lead one to erroneously suspect its breakage.

In human medicine, the Hawkins test is described as a specific test for evaluating the supraspinatus tendon. The patient first flexes her/his arm by 90 degrees, then flexes her/his elbow by 90 degrees and intrarotates his/her shoulder. Pain indicates that the test is positive, and the pain is due to the compression of the biceps tendon by the tendon of the supraspinatus [15]. In our experience, we observed a progressive increase in the operator’s capacity to identify the supraspinatus tendon through palpation during the clinical examination and to evoke pain by exerting digital pressure in its insertion on the major tubercle. However, it is difficult to differentiate the pain evoked by the compression of the supraspinatus tendon from pain evoked during the execution of the biceps test in the English-speaking literature. We can argue that many lesions in the past were considered biceps tendinopathy because of a positive response to biceps tendon test in the English-speaking literature. A great number of patients included in the study presented with muscle atrophy, and this can help to identify the insertion of the tendon. We must emphasize that the significance of tendon palpation for correctly formulating [33] the diagnosis has never been evaluated in the literature and we do not know its sensitivity and specificity.

In adult patients referred for shoulder lameness, it is always necessary to carry out a orthogonal comparative radiographic study even if the majority of shoulder pathologies involve the tenoligamentous structures [7,34]. A radiographic study can exclude some pathologies such as tendon avulsion of the biceps brachii muscle or the mineralized form of supraspinatus tendinopathy. In dogs affected by the nonmineralized form of supraspinatus tendinopathy, the only visible alteration is the presence of osteophytes in the bicipital groove visible in the skyline projection [7]. Some authors attribute the presence of osteophytes to arthritis determined by irritation of the biceps tendon and its sheath, and by the compression applied by a thickened supraspinatus tendon [7].

In our study, we examined the synovial fluid in only 16% of the cases, but we considered this to be a mistake during the diagnostic process. It is advisable to include a cytologic examination of the synovial fluid in the diagnostic work out in order to rule out pathologies such as neutrophilic arthrosynovitis before considering MRI of the shoulder.

In the dogs included in this study, given the negative results of the radiographic and synovial liquid exams, we advanced nonmineralized tendinopathy of the supraspinatus as a probable diagnosis, and we informed the owners that this diagnosis could be confirmed through ultrasound examination or magnetic resonance imaging [7]. With the possibility of choosing between ultrasound and MRI, the authors always prefer MRI. The most common alterations highlighted by an ultrasound examination are the enlargement of the tendon with or without axial deviation of the biceps tendon and inhomogeneities in the tendon [1,33]. However, the alterations highlighted by the ultrasound exam are subjective and their interpretation is strongly influenced by the operator’s experience [7]. MRI is a powerful diagnostic modality, commonly used for diagnoses of human musculoskeletal diseases and providing information on the intra- and extra-articular structures simultaneously with high soft tissue contrast, and high-resolution and multiplanar imaging capabilities [35]. Compared with ultrasound, it allows the evaluation of teno-ligamentous structures such as the subscapular muscle and collateral ligaments that are not fully reached by ultrasound. This can provide the surgeons with more information on the status of the shoulder joint compared with ultrasound alone. For this reason, we prefer to advise for MRI, which represents the gold standard for diagnosing extra-articular pathologies of the shoulder [1,34]. Despite this, it is the opinion of the authors that, in further studies, a comparison between MRI and ultrasound findings could be very useful in order to better define diagnostic criteria of the nonmineralized supraspinatus tendinopathy.

The common alteration revealed by MRI in patients affected by nonmineralized tendinopathy is an enlargement and a marked increase in the signal in the insertion area of the supraspinatus tendon above the greater tubercle of the humerus in the STIR sequences. In a recent study, it was noticed that pathologic tendons, in addition to an alteration in the signal, also present an increase in volume [22]. In our population, only 3 cases out of 27 had an increase in the volume in addition to an alteration in the signal.

The alterations indicated by MRI must always be associated with the clinical data.

In our case studies, it was interesting to note that there was some discrepancy in the evaluation of the shoulder joint when we compared the findings of MRI and arthroscopy in the biceps and subscapularis tendons and the medial collateral ligament. Regarding the biceps tendon, MRI highlighted an ectasia in 10 out of 27 cases that was not confirmed by arthroscopy. Arthroscopy indicated a partial rupture and laxity of the subscapularis tendon in 10 out of 27 cases that was not highlighted in 9 out of 27 cases through MRI. Finally, arthroscopy highlighted laxity of the medial collateral ligament in 3 out of 27 cases that was not displayed in MRI in 0 out of 3 cases. For this reason, as arthroscopy is a minimally invasive technique which requires a short prolongation of the patient’s anesthesia with a very low incidence of complications, it is considered useful to always carry it out before surgery, as it often provides useful information for understanding the pathology and the prognosis of the patient. As well as ultrasound, evaluation by MRI imaging presents minimal variability caused by the operator’s experience. Having a good evaluation of the means of intra-articular stability is also fundamental for evaluating the prognosis and for the functional recovery of the patients treated surgically. In our case studies, 37% of cases had a subscapularis tendon injury that can lead to the persistence of lameness in the post-operative period, which must be discussed with the owners.

Dogs at our institution affected by supraspinatus tendinopathy had a mechanical alteration of the gait that led them to abduction of the limb. We argue that this gait alteration may be the cause of the torn subscapularis tendon or the partial rupture of the medial collateral ligament, which were often observed in our population; otherwise, these lesions can be considered to be concomitant to the nonmineralized supraspinatus tendinopathy. It is reasonable to consider those lesions as of lesser regard because the lameness often resolves after the treatment of the supraspinatus tendinopathy alone. Dogs included in this study that presented a concomitant injury to the subscapularis tendon or medial collateral ligament were treated in the post-operative period [36] using dedicated hobbles. The use of hobbles was recommended for at least 3 months.

In our cases, slower functional recovery with a protracted lameness was reported in patients affected by concomitant supraspinatus and biceps brachii tendinopathy. It is supposed that, if not treated, the biceps brachii tendinopathy might have a consistent influence on the post-operative lameness score compared with the other concomitant lesions observed by means of arthroscopy or by MRI.

In the literature, the craniomedial surgical approach has been reported for executing splitting of the supraspinatus tendon [7,37]. In our case studies, we decided to adopt the dorsal recumbency position in the patients, which allowed the operators to perform a very small surgical approach and to limit the dissection of soft tissues. The most frequent complication found in our cases was seroma, which occurred in 18% of cases. To avoid this complication, we adopted the quilting suturing technique [38].

The execution of tendon splitting is easy and involves making longitudinal release incisions at the level of the muscle–tendon junction [7,37]. This technique induces a reduction in the intra-tendinous pressure, an increase in the critical zone’s vascularization and a reduction in the healing time of the tendon [7]. However, it is sometimes difficult to macroscopically identify the pathological area and the incisions may remain localized in the tendinous portion. If lameness persists in the post-operative period, it may be secondary to the execution of incisions in a nonpathological portion of the tendon or to the formation of excessive scar tissue, as we observed in one case [19].

The histopathologic examination of collected samples highlighted severe myxomatous degeneration, with the remaining collagen separated by a palely basophilic myxoid matrix and edema. Pownder et al. [23] suggested that histologic evidence of mucinous or chondroid degeneration in focal biopsy specimens should be interpreted cautiously because the identification of well-organized structures during examination of the entire tendon might refute such a diagnosis. In our study all samples were obtained from the area of maximum thickness at the musculotendinous junction where mineralization foci are commonly localized in the mineralized form of the pathology. This portion of the tendon was characterized by a thickened appearance and, during surgical splitting, was characterized by a “crunchy” sound. Control specimens of normal tendons were unavailable for comparison purposes; considering those findings it is possible that for some cases to have been incorrectly interpreted as evidence of mucinous or chondroid degeneration indicative of maladaptive or pathological lesions. More studies are still required in order to better define the histopathological features of the non-mineralized form of supraspinatus tendinopathy.

The visual and histological examination of some tendinous portions highlighted the presence of mineralized foci in the tendon samples of subjects affected by the nonmineralized form of the pathology. This may support the theory that the nonmineralized tendinopathy is an initial phase of the more common mineralized form, so we can suppose that in dogs with the nonmineralized form, the mineralized form could develop over time if surgery is not carried out. Some authors affirm that degeneration of the tendon can follow a cycle of degeneration and regeneration. Interruption of this cycle can cause the progressive formation of mineralized foci and the development of clinical signs [18,39,40,41].

In 21 out of 27 (77%) patients, there was a marked improvement in lameness after the removal of stitches 10 days after surgery. The 30-day follow-up was satisfactory in 81% of cases, but in only one case, we decided, given the persistence of marked lameness, in favor of surgical re-intervention. During the surgery, fibrosis was observed in the portion where the tendon splitting had been performed, so we executed a partial tenectomy of the tendon, which led to rapid remission of the symptoms.

To our knowledge, long-term follow-up of patients affected by the mineralized or nonmineralized forms of supraspinatus tendinopathy has never been reported in the literature. Anecdotally, according to some surgeons, it would be better to perform a complete tenectomy to avoid the risk of recurrence. In our study, we performed a long-term follow-up between 6 and 9 months after the surgery. The long-term follow-up included a telephone interview, and clinical and radiological examinations. In 85% of patients, the result was considered to be satisfactory by the owners and, in 81% of the cases, lameness was no longer reported. These data support the efficacy of the tendon splitting technique for the treatment of nonmineralized supraspinatus tendinopathy.

Furthermore, the long-term follow-up allowed us to highlight how although the MRI had shown an alteration in the STIR signal in the contralateral limb in 11% of cases, none of these patients ever became symptomatic. This shows how the alterations found by MRI must always be supported by clinical data to have value; therefore, it is not necessary to perform surgery for preventive purposes in the absence of symptoms.

For cases in which it was possible to execute the force plate examination in the long-term clinical examination, we observed a low symmetry index, with an average value of 1.58. As demonstrated by Grandjean e Fanchon, this value, being less than 3.2, can be considered physiological [30,42,43].

This study has the implicit limitations of retrospective studies. One limit is related to the fact that the information was extracted retrospectively from the medical records. Another limitation is related to the impossibility of executing a long-term clinical follow-up in some patients; therefore, the data obtained have a limited value because they are based on the subjective judgement of the owners. However, it is believed that the owners are able to effectively evaluate their dog in the home environment, which is why the owner’s opinion should not be underestimated as part of the evaluation of the results [44].

## 5. Conclusions

In conclusion, we can affirm that supraspinatus tendinopathy is a common pathology in medium to large breeds of dog weighing more than 20 kg. The treatment of tendinopathy is a quick and easy procedure to perform, with no intra-operative complications and a low incidence of minor (18%) and major (4%) complications. In order to avoid complications and the necessity of revision surgery, it is considered fundamental to make the incisions of the splitting procedure exactly between the muscular and tendinous junctions of the supraspinatus muscle, taking care to avoid any damage to the tendon of the biceps brachii muscle. The dogs enrolled in the study showed a satisfactory functional recovery. The resolution of lameness was achieved in 80% of the cases without any further lameness episodes in the long-term follow-up.

## Figures and Tables

**Figure 1 animals-13-00592-f001:**
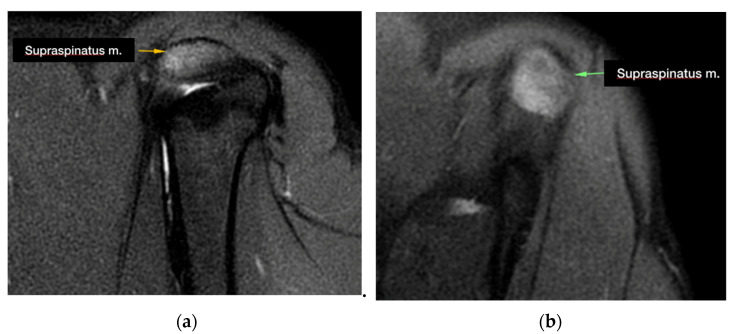
Supraspinatus tendon appearance during MRI imaging acquisition: (**a**) hyperintense signal at the level of the insertion of the supraspinatus tendon in the STIR images (sagittal plane); (**b**) hyperintense signal at the level of the insertion of the supraspinatus tendon in the STIR images (transverse plane).

**Figure 2 animals-13-00592-f002:**
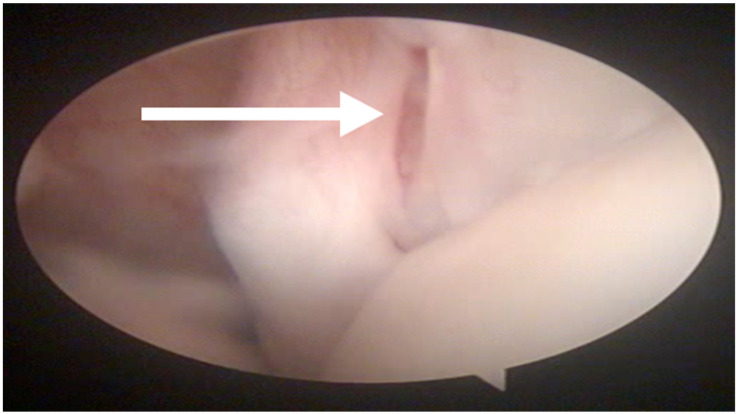
Arthroscopic evaluation of the tendon of the sub-scapularis muscle. A partial rupture is indicated by the white arrow.

**Figure 3 animals-13-00592-f003:**
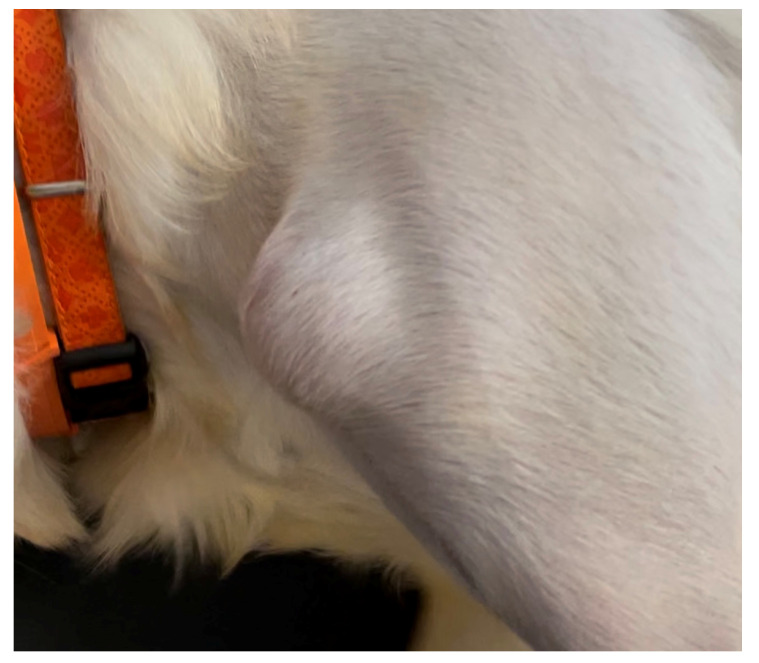
Seroma formation on the surgical site 20 days after surgery.

**Figure 4 animals-13-00592-f004:**
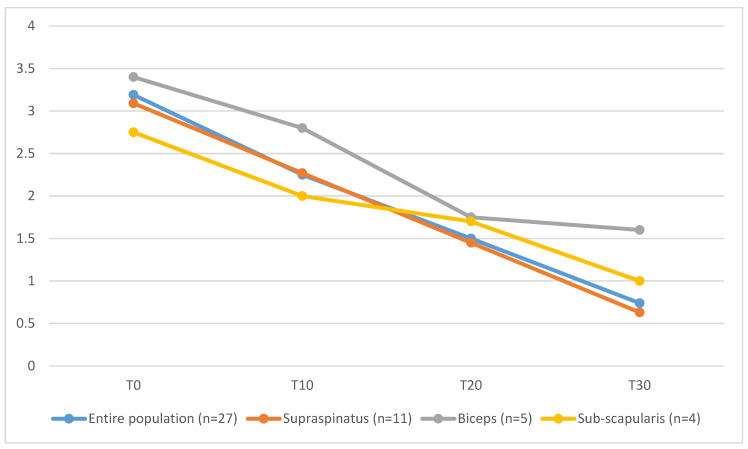
Mean lameness scores values of the entire population and divided into 3 sub-groups based on others concomitant lesions investigated during the arthroscopic evaluation are reported for each time frame. Blue line: values are reported for the entire population. Orange line: subjects affected by ST without any other concomitant lesions. Grey line: subjects affected by ST and biceps brachii lesions. Yellow line: subjects affected by ST and sub-scapularis tendon lesions.

## Data Availability

Not applicable.

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
