# Peer review of "Surgical Treatment of Nonmineralized Supraspinatus Tendinopathy in Dogs: A Retrospective Long-Term Follow-Up"

_animals, 2023, doi:10.3390/ani13040592_

Round 1

Reviewer 1 Report

Dear Authors,

the paper aimed to retrospectively evaluate the clinical outcome of dogs affected by nonmineralized supraspinatus tendinopathy surgically treated.

The study looks well organized, with interesting efforts regarding the comparison between the arthroscopic/MRI findings and the force plate evaluation. 

Please find below my few suggestions:

Abstract

Ln 27-28: in dogs?

Ln 30:  and the shot…. Please delete

Ln 41: please specify the surgical treatment you performed

Introduction

Ln 53-54: please add references

Ln 56: any hypthosesis about the left hindlimb?

MM

Ln 81-82: did you include mono/bilateral condition? Just mono? Please clarify

Ln 114: please delete all the highlighted words

Ln 141: an incision was performed.. in which direction?

Ln 143-144: did you scale your inspection/palpation of the tendon?

Ln 146-147: too much time you used “closed”, please change

Ln 153: did you give any antibiotics in the peri and post period? Please clarify

Ln 164: any potential intraop complications?

Results

Ln 227: delete “median lameness was 3”

Ln 232: please better define the OA changes, regarding the position and grade (slight, mild, moderate, severe..)

Ln 254: was the surgical procedure always performed by the same operator?

Discussion

Ln 418: do you suggest shoulder ultrasonography in an ideal work up? Ord o you suggest directly MRI?

In your study population one dog had tenectomy as revision surgery. Why do you think the splitting was unsuccessful? Please discuss this item.

Thank you

Reviewer 2 Report

Dear Authors

I reviewed the manuscript entitled "Surgical treatment of nonmineralized supraspinatus tendinopathy in dogs: A retrospective long-term follow-up" by Piras et al. The manuscript is interesting since nonmineralized  tendinopathy of the Supraspinatus tendon is seldom reported. 

I have some concerns on the materials and methods section, since diagnostic criteria are not clearly described (see specific comments), results are not consistent with materials and methods in some way and discussion is too long and confusing in my opinion. Therefore I recommend major revision.

Specific comments:

Abstract

line 30 : should be short instead of shot

Materials and methods:

Imaging: Please clarify the MRI criteria of ST tendinopathy. MRI findings of ST tendinopathy are inconstant. Pownder et al (Magnetic resonance imaging and histologic features of the supraspinatus muscle tendon in nonlame dogs, AJVR 2018) found that histologically normal supraspinatus tendons have a trilaminar appearance with a thick central substance that is hyperintense on MRI images".

Moreover Scheffer et al (Veterinary surgery 2006, Magnetic Resonance Imaging of the Canine Shoulder: An Anatomic Study) stated that " Tendons and ligaments had a wide variety of signal intensity from the hyperintense supraspinatus tendon to the hypointense subscapularis tendon".

I suggest that the presence of other MRI criteria such as edema at the myotendinous junction or greater tubercle of the humerus and/or evidence of tendon avulsion and other evidence of joint degeneration can be used to increase the index of suspicion for ST tendinopathy. 

Surgical technique: Did the dogs undergo physiotherapy in the postoperative period?

Results

Findings of the Orthopaedic examination: please add some information on the history (duration of the lameness and previous treatment)

Lines 234-236 The Authors stated: "hyperintense signal localized to the insertion of the supraspinatus muscle was found in all of the examined shoulders in the STIR images; in the T2-weighted images, this was noted in 2 out of 27 dogs, while a nonhomogeneous signal was noted in the FSE-T2 images in one dog". Please explain these findings. Was magic angle artifact taken into account?

Lines 259-262. "The arthroscopic and MRI findings were in agreement in 10 out of 27 shoulders without abnormalities, in 14 out of 27 shoulders regarding the biceps brachii, in 18 out of 27 regarding the subscapularis muscle, in 24 out of 27 in regard to the medial glenohumeral ligament and in 27 out of 27 about the lateral glenohumeral ligament". Please explain these findings more thoroughly, as they are unclear to me.

Histopathological Findings: please explain the site of ST biopsy. Pownder et al suggest that histologic evidence of mucinous or chondroid degeneration in focal biopsy specimens should be interpreted cautiously because the identification of well-organized structures during examination of the entire tendon might refute such a diagnosis. If those specimens were obtained from the central substance of the tendon and appropriate control specimens were unavailable for comparison purposes, those findings may have been incorrectly interpreted as evidence of mucinous or chondroid degeneration indicative of maladaptive or pathological lesions. 

Discussion

Discussion section is very long and add several confusing information, not described in materials and methods and result sections. Please, rearrange the section on the light of the aim of the study.

line 358-361: These information were not presented in results.

line 373-374:" In our experience, this type of lameness is very different from lameness secondary to other shoulder pathologies, such as biceps brachii tendinopathy": This sentence is unclear.

line 376: During the clinical evaluation, the biceps test (in the German-speaking literature) was performed, and it was wrongly judged to be positive in 3 out of 27 cases: sentence unclear

line 408: In our study, we examined the synovial fluid in only 16% of the cases: this finding was not described in materials and methods and in results.

line 411: Examination of the synovial fluid demonstrated neutrophilic arthrosynovitis. In such cases, an expensive imaging technique such as MRI could be avoided: sentence unclear.

line 416:" In the dogs included in this study, given the negative results of the radiographic and synovial liquid exams, we advanced nonmineralized tendinopathy of the supraspinatus  as a probable diagnosis, and we informed the owners that this diagnosis could be confirmed through ultrasound examination or magnetic resonance imaging [7]. With the possibility of choosing between ultrasound and MRI, the authors always prefer MRI": why not using both techniques? 

Table 1: is confusing and unclear to me. 

Round 2

Reviewer 2 Report

Dear Authors 

I reviewed the manuscript entitled " Surgical treatment of nonmineralized supraspinatus tendinopathy in dogs: A retrospective long-term follow-up". The manuscript is improved very much. 

I have only a minor comment in the discussion section (line 444) on shoulder ultrasound examination. It would be very interesting a comparison between MRI and ultrasonographic findings in further studies on nonmineralized supraspinatos tendinopathy.
